# Incidence and Mortality of Emergency Patients Transported by Emergency Medical Service Personnel during the Novel Corona Virus Pandemic in Osaka Prefecture, Japan: A Population-Based Study

**DOI:** 10.3390/jcm10235662

**Published:** 2021-11-30

**Authors:** Yusuke Katayama, Kenta Tanaka, Tetsuhisa Kitamura, Taro Takeuchi, Shota Nakao, Masahiko Nitta, Taku Iwami, Satoshi Fujimi, Toshifumi Uejima, Yuuji Miyamoto, Takehiko Baba, Yasumitsu Mizobata, Yasuyuki Kuwagata, Takeshi Shimazu, Tetsuya Matsuoka

**Affiliations:** 1Department of Traumatology and Acute Critical Medicine, Osaka University Graduate School of Medicine, Suita 565-0871, Japan; shimazu@hp-emerg.med.osaka-u.ac.jp; 2Department of Social and Environmental Medicine, Division of Environmental Medicine and Population Sciences, Osaka University Graduate School of Medicine, Suita 565-0871, Japan; tanaken.0414@gmail.com (K.T.); lucky_unatan@yahoo.co.jp (T.K.); tarogauss106072@gmail.com (T.T.); 3Senshu Trauma and Critical Care Center, Rinku General Medical Center, Izumisano 598-8577, Japan; shotaan24@gmail.com (S.N.); t-matsuoka@rgmc.izumisano.osaka.jp (T.M.); 4Department of Emergency Medicine, Osaka Medical and Pharmaceutical University, Takatsuki 569-8686, Japan; nittam@osaka-m.ac.jp; 5Kyoto University Health Service, Kyoto 606-8501, Japan; Iwami.taku.8w@kyoto-u.ac.jp; 6Division of Trauma and Surgical Critical Care, Osaka General Medical Center, Osaka 558-8558, Japan; sfujimi40@nifty.com; 7Emergency Care Center, Kindai University Hospital, Osaka-Sayama 589-8511, Japan; uejiman@gmail.com; 8Otemae Hospital, Osaka 540-0008, Japan; ymiyamoto@otemae.gr.jp; 9Baba Memorial Hospital, Sakai 592-8341, Japan; pegasushoujinhonbu@yahoo.co.jp; 10Department of Traumatology and Critical Care Medicine, Osaka City University Graduate School of Medicine, Osaka 558-8585, Japan; mizobata@med.osaka-cu.ac.jp; 11Department of Emergency and Critical Care Medicine, Kansai Medical University, Hirakata 573-1010, Japan; kuwagata@hirakata.kmu.ac.jp

**Keywords:** COVID-19, emergency medical service, ambulances, incidence, mortality, epidemiology

## Abstract

Although the COVID-19 pandemic affects the emergency medical service (EMS) system, little is known about the impact of the COVID-19 pandemic on the prognosis of emergency patients. This study aimed to reveal the impact of the COVID-19 pandemic on the EMS system and patient outcomes. We included patients transported by ambulance who were registered in a population-based registry of patients transported by ambulance. The endpoints of this study were the incident number of patients transported by ambulance each month and the number of deaths among these patients admitted to hospital each month. The incidence rate ratio (IRR) and 95% confidence interval (CI) using a Poisson regression model with the year 2019 as the reference were calculated. A total of 500,194 patients were transported in 2019, whereas 443,321 patients were transported in 2020, indicating a significant decrease in the number of emergency patients transported by ambulance (IRR: 0.89, 95% CI: 0.88–0.89). The number of deaths of emergency patients admitted to hospital was 11,931 in 2019 and remained unchanged at 11,963 in 2020 (IRR: 1.00, 95% CI: 0.98–1.03). The incidence of emergency patients transported by ambulance decreased during the COVID-19 pandemic in 2020, but the mortality of emergency patients admitted to hospital did not change in this study.

## 1. Introduction

Outbreaks of infection by the novel corona virus (COVID-19), which was confirmed in Wuhan, China in December 2019, have spread not only in China but also around the world. In Japan, the number of patients with COVID-19 was about 740,000 on 31 May 2021 [1]. The characteristics of COVID-19 are that some of its symptoms, such as fever, cough, sore throat, and general malaise, are common with other upper respiratory tract infections, and some patients are asymptomatic [2]. However, 20% of COVID-19 patients are severely affected and admitted to hospital, and a lower but not negligible rate (3–4%) also need intensive management in the ICU, for their acute respiratory failure, by intubation and mechanical ventilation [3].

As the number of patients with COVID-19 increased, especially in Europe and the United States, the number of health care workers infected with COVID-19 also increased, placing aspects of the health care system, such as emergency medicine and intensive care, into a worldwide state of crisis [4]. The health care system in Japan is funded by public health insurance, and the emergency medical service (EMS) system, which handles all ambulance calls, is a free public service [5]. However, the impact of the COVID-19 pandemic on the EMS system has not been fully revealed, and little is known about the impact of the COVID-19 pandemic on the prognosis of emergency patients.

Osaka Prefecture is the largest metropolitan area in western Japan, with a population of 8.8 million. The annual number of ambulance calls is about 500,000 in this area and that of patients transported to hospital by ambulance is about 200,000 [6]. After the first patient in Osaka Prefecture was confirmed to have COVID-19 on 23 January 2020, the cumulative number of patients with COVID-19 in the prefecture rose to 1732 by 31 May 2020, which was considered the first surge of COVID-19 [7]. We previously revealed the characteristics and outcome of patients with COVID-19 in Osaka Prefecture [7]. Those patients in Osaka Prefecture suspected of having COVID-19 based on their medical and travel history were transferred to a hospital that specializes in the management of COVID-19 for PCR testing. When a COVID-19 outbreak was reported in places such as bars and live music venues, the staff in each public health centre in charge followed up on the people involved, and data on the individuals with positive PCR test results were collected to determine whether they were asymptomatic. All patients with positive PCR test results for COVID-19 were reported to the public health centres in accordance with the Infectious Disease Control Law [8]. In Osaka Prefecture, the first patient with COVID-19 was identified on 23 January 2020, and by 31 December 2020, 466,416 PCR tests had been conducted and the number of patients with COVID-19 was 29,999 [9]. In Japan, due to an increase in the number of patients with COVID-19, the Japanese government declared a state of emergency based on the law on 7 April 2020. At that time, we revealed the influence of the COVID-19 pandemic on the EMS system in Osaka City [10]. The goals of this investigation were to determine the impact of the COVID-19 pandemic on the incident number of emergency patients transported by ambulance (emergency patients) and the number of deaths of emergency patients admitted to hospital.

## 2. Materials and Methods

### 2.1. Study Design and Settings

This was a retrospective descriptive study with a study period from 1 January 2019 to 31 December 2020. All data about patients who were transported by ambulance from ambulance call to hospital discharge were entered into the ORION (Osaka Emergency Information Research Intelligent Operation Network) system. Information on the system configuration of ORION was previously described in detail [6,11]. ORION data are considered administrative records, and the ORION data are anonymized without specific personal data, such as patient name, date of birth, and address. Therefore, the requirement of obtaining patient informed consent was waived. This study was approved by the Ethics Committee of Osaka University Graduate School of Medicine (approval no. 15003).

### 2.2. Setting and Selection of Patients

In 2019, 8,823,452 people lived in the 1905 km^2^ area of Osaka Prefecture [12]. Of that population, 4,235,996 people (48.0%) were male and 2,382,016 people (27.0%) were elderly, aged 65 years old or more. We included patients transported by ambulance whose cleaned data were recorded in the ORION system. Therefore, we excluded patients who were not registered in the ORION system or who had missing data.

### 2.3. Outcomes

The primary endpoints of this study were the incident number of patients transported by ambulance in each month of the study period and the number of deaths of emergency patients admitted to hospital in each month. In this study, patients who died in the emergency department were excluded from the outcome.

### 2.4. Measurements

The ORION system checks for errors in the input in-hospital data, and the staff of each emergency hospital can correct them, if necessary. Through these tasks, cell phone app data, ambulance records, and the in-hospital data such as diagnosis and prognosis can be comprehensively registered for each patient transported by an ambulance. The registered data are cleaned by the Working Group to analyse the emergency medical care system in Osaka Prefecture. Among the collected and cleaned data, we excluded inconsistent data that did not contain all of the cell phone app data, ambulance records, and in-hospital data such as diagnosis and prognosis. In addition, we also excluded patients whose sex as registered by the fire department did not match that registered by the hospital or whose sex identifier was missing. We also excluded patients whose age input by the fire department and that by the hospital differed by 3 years or more. When this difference was present, we defined the age input by the hospital as the patient’s true age [5].

### 2.5. Data Analysis

First, we calculated the number of patients transported by ambulance by reason for ambulance call on a monthly basis from January to December 2020. As a control, we calculated the same data on a monthly basis from January to December 2020. Reason for ambulance call was divided into ‘fire accident’, ‘natural disaster’, ‘water accident’, ‘traffic accident involving car, ship, or aircraft’, ‘injury, poisoning, and disease due to industrial accident’, ‘disease and injury due to sports’, ‘other injury’, ‘trauma due to assault’, ‘acute disease’, ‘interhospital transport’, and ‘others’ [6,11]. To evaluate the impact of the COVID-19 pandemic on the EMS system, we calculated the incidence rate of the number of emergency patients. We also calculated the incidence rate ratio (IRR) and its 95% confidence interval (CI) using a Poisson regression model with the year 2019 as control year. We categorized the patients by age group (children (0–19 years old), adult (20–64 years old), and elderly (65 years old and over)) and also calculated their respective IRR and 95% CI values. Next, we calculated the number of deaths of emergency patients admitted to hospital by reason for ambulance call in each month and similarly calculated the IRR and its 95% CI values. The offset for calculating the IRR was set to the population of Osaka Prefecture in 2019 (8,823,452 people) [12]. The death of emergency patients admitted to hospital was defined from the outcome at 21 days after hospital admission. In addition, in a subgroup analysis, we selected the patients transported by ambulance whose reason for ambulance call was ‘acute disease’ and similarly calculated the IRR and 95% CI values. Statistical analyses were performed using STATA version 16.0 MP software (StataCorp LP, College Station, TX, USA). This manuscript was written based on the STROBE statement to assess the reporting of cohort and cross-sectional studies [13]. All methods in this study have been carried out in accordance with the declaration of Helsinki.

## 3. Results

The total number of patients registered in ORION was 512,054 in 2019, of which 500,194 (97.7%) were eligible for analysis after excluding cases with missing data. In addition, the total number of patients registered in ORION was 451,524 in 2020, of which 443,321 (98.2%) were eligible for analysis after excluding cases with missing data. Among the 443,321 patients registered in the ORION registry from January to December 2020, 193,060 patients were hospitalized, and 11,963 patients were dead at 21 days after hospital admission. In contrast, among the 500,194 patients registered in the ORION system from January to December 2019, 203,889 patients were hospitalized, and 11,931 patients were dead at 21 days after hospital admission.

### 3.1. Incidence Analyses by Reason of Ambulance Call

Table 1 shows the number of emergency patients and the IRR (95% CI) in each month by the reason for ambulance call during the study period. The number of emergency patients from January to December 2020 (*n* = 443,321) was significantly decreased from that transported from January to December 2019 (*n* = 500,194) (IRR: 0.89, 95% CI: 0.88–0.89). The most common reason for an ambulance call was ‘acute disease’ for 340,655 patients in 2019 and 300,502 patients in 2020. During the study period, the reasons for an ambulance call for which the number of emergency patients decreased were ‘traffic accident involving car, ship, or aircraft’ (IRR: 0.86, 95% CI: 0.85–0.87), ‘injury, poisoning, and disease due to industrial accident’ (IRR: 0.82, 95% CI: 0.79–0.86), ‘disease and injury due to sport’ (IRR: 0.57, 95% CI: 0.53–0.60), ‘other injury’ (IRR: 0.92, 95% CI: 0.91–0.93), ‘trauma due to assault’ (IRR: 0.88, 95% CI: 0.84–0.93), ‘acute disease’ (IRR: 0.88, 95% CI: 0.88–0.89), and ‘interhospital transport’ (IRR: 0.90, 95% CI: 0.89–0.91). By month, the greatest decrease in the number of emergency patients was in April (IRR: 0.78, 95% CI: 0.76–0.79), followed by May (IRR: 0.79, 95% CI: 0.78–0.80).

Table 2 shows the number of emergency patients and the IRR (95% CI) in each month by the age groups during the study period. In the subgroup analysis by age group, the number of emergency patients decreased among children during the study period (IRR: 0.68, 95% CI: 0.67–0.69). However, for adults and the elderly, the number of emergency patients decreased after March 2020 compared to that in 2019.

### 3.2. Mortality Analyses by Reason of Ambulance Call

Table 3 shows the number of deaths of emergency patients admitted to hospital and the IRR (95% CI) in each month by the reason for ambulance call. The number of deaths of emergency patients admitted to hospital was 11,931 in 2019 and remained essentially unchanged at 11,963 in 2020 (IRR: 1.00, 95% CI: 0.98–1.03). There was no statistically significant change in the number of deaths of emergency patients admitted to hospital for each reason for an ambulance call between 2019 and 2020, and no statistically significant differences were identified between 2019 and 2020 for each month.

Table 4 shows the number of deaths of emergency patients admitted to hospital and the IRR (95% CI) in each month by age groups. In subgroup analysis by age group, there was no increase of the number of deaths of emergency patients admitted to hospital among children (IRR: 0.81, 95% CI: 0.54–1.21), adults (IRR: 0.98, 95% CI: 0.91–1.05), and the elderly (IRR: 1.01, 95% CI: 0.98–1.04).

### 3.3. Subgroup Analyses by Age Groups among Patients with Acute Disease

Table 5 shows the number of emergency patients due to acute disease by age group and the IRR (95% CI) for each month during the study period. The number of paediatric patients transported by ambulance during the study period significantly decreased (30,961 patients in 2019 vs. 18,929 patients in 2020; IRR: 0.61, 95% CI: 0.60–0.62). The number of adult patients transported by ambulance also significantly decreased (107,634 patients in 2019 vs. 95,355 patients in 2020; IRR: 0.89, 95% CI: 0.88–0.89), as did that of the elderly patients transported by ambulance (202,620 patients in 2019 vs. 186,218 patients in 2020; IRR: 0.92, 95% CI: 0.92–0.93).

Table 6 shows the number of deaths of emergency patients admitted to hospital due to acute disease by age group and IRR (95% CI) for each month. The number of deaths among emergency paediatric patients admitted to hospital due to acute disease was 26 in 2019 and 25 in 2020 (IRR: 0.96, 95% CI: 0.53–1.73). The number of deaths among emergency adult patients admitted to hospital due to acute disease was 1210 in 2019 and 1171 in 2020 (IRR: 0.97, 95% CI: 0.89–1.05), and that among emergency elderly patients admitted to hospital due to acute disease was 8591 in 2019 and 8660 in 2020 (IRR: 1.01, 95% CI: 0.98–1.04). No statistically significant differences were identified between 2019 and 2020 for each month or by age group.

## 4. Discussion

In this study, we used data from a large population-based patient registry to determine the number of emergency patients and the number of deaths among these patients admitted to hospital in the COVID-19 pandemic during 2020 in Osaka Prefecture. Although the number of emergency patients decreased in 2020 compared with 2019, the number of deaths among the emergency patients admitted to hospital in 2020 was similar to that in 2019. The results of this study, which used population-based data to reveal the impact of an emerging infectious disease pandemic on the EMS system, could be useful to plan health care systems and policies.

The number of emergency patients decreased in 2020 compared with 2019, especially in April, May, and December. As well, the number of emergency patients due to acute disease as the reason for the ambulance call also decreased, especially in April, May, and December. A previous study in Venice, northern Italy, comparing the number of ambulance dispatches in 2019 and 2020, found that the COVID-19 pandemic reduced the number of ambulance dispatches in 2020 [14]. It was also reported that the number of emergency department visits decreased during the severe acute respiratory syndrome (SARS) pandemic that spread in 2003 [15,16,17,18,19]. Thus, when an infectious disease spreads throughout a city or society, the number of emergency department visits may decrease as a result of people buying medicines from pharmacies for their own care and refraining from visiting the emergency department. In contrast, in Seine-Saint-Denis, which is a French department bordering Paris to the northeast and is a part of Greater Paris, Lapostolle et al. reported that the COVID-19 pandemic increased the number of calls for the Service d’Aide Medicale Urgente (SAMU) and the number of emergency department visits compared to the average of the previous five years [20]. The SAMU in France provides several medical services such as medical advice and hospital transfer by a non-emergency transport ambulance. Contrastingly, the only service provided by the EMS system in Japan is ambulance dispatch, and the differences in services provided by the SAMU in France versus the EMS system in Japan may have affected the difference in results. Further, Saberian et al. reported an increase in the number of EMS calls and ambulance dispatches after the first COVID-19 patient was identified on 18 February 2020 in Tehran, Iran [21]. The EMS system in Iran is similar to that in Japan in that the EMS personnel evaluate the patient at the scene and, if necessary, transport the patient to a hospital. The difference of results between the study in Japan and that in Iran, which operates a similar EMS system, may be due to the fact that Japanese people who used to call an ambulance even in cases not necessarily requiring an ambulance are now discouraged from visiting hospitals and clinics due to the risk of COVID-19.

The number of emergency patients due to sports injuries, industrial accidents, and traffic accidents also decreased in 2020 compared to 2019. In Japan, the Japanese government requested temporary closures of elementary, junior high, and high schools on 2 March 2020 [22], and the temporary closure of these schools continued until 31 May 2020 in Osaka Prefecture. In addition, many sports gyms have refrained from operating as a result of COVID-19 outbreaks in some of these gyms. As a result of this reduction in opportunities for sports in schools and gyms, the number of emergency patients due to sports injuries would likely have decreased. In Japan, although no explicit lockdown measures were taken by the government, the number of emergency patients due to traffic accidents and industrial accidents may have also decreased because of the slowdown in socioeconomic activity due to the voluntary restraint of various companies. Subgroup analyses by age group showed a decrease in patients transported by ambulance among children starting in January and a decrease in patients transported by ambulance among adults and the elderly after March. This result may be due to parents being less likely to visit the emergency department due to vigilance against an unknown infectious disease. In addition, as a result of school closures, they may not have visited emergency departments as a result of fewer cases of seasonal influenza in their children.

There was no change in the number of deaths of emergency patients admitted to hospital in 2020 compared with 2019. There were also no differences in the number of deaths of emergency patients admitted to hospital in the analyses by reason for ambulance call or by age group. Indeed, several previous studies have reported that COVID-19 outbreaks have reduced emergency patients due to influenza and mortality due to other infectious diseases [23,24]. On the other hand, there were concerns that other acute illnesses might affect the prognosis of emergency elderly patients due to an increase in demand for medical care. However, no impact on their prognosis was identified in this study because the health care system and EMS system functioned effectively for the community as a whole. To maintain the level of medical treatment in future surges of the COVID-19 pandemic and other infectious disease pandemics, it will be necessary to establish a medical and health care system with a clear role for medical institutions.

This study has several limitations. First, although all fire departments and emergency medical institutions in Osaka Prefecture registered ambulance records and patient data in the ORION registry, the prognosis of patients transported to medical institutions outside Osaka Prefecture or by fire departments outside Osaka Prefecture is unknown. Second, no information was available on the detailed treatment of the patients in hospital that would have affected death after hospital admission. Third, although this study was analysed by reason for ambulance call, a detailed analysis of the impact of the COVID-19 pandemic on the EMS system by disease, such as out-of-hospital cardiac arrest, acute coronary syndrome, and pneumonia, will be performed and reported in the near future. Fourth, as we included the emergency patients in this study, the impact of the COVID-19 pandemic on all causes of death in Osaka was unknown. Fifth, we did not include the deaths in the emergency department in this study. Many of the patients who died in the emergency department were the patients with out-of-hospital cardiopulmonary arrest. Prehospital factors such as bystander cardiopulmonary resuscitation can affect the outcomes of patients with out-of-hospital cardiopulmonary arrest. Therefore, we did not include these patients in this study.

## 5. Conclusions

In Osaka Prefecture, Japan, the incidence of emergency patients transported by ambulance decreased during the COVID-19 pandemic in 2020, but the mortality of emergency patients admitted to hospital did not change. The impact of the COVID-19 pandemic on the EMS system will need to be monitored over the long term.

## Figures and Tables

**Table 1 jcm-10-05662-t001:** The number of emergency patients registered in the Osaka Emergency Information Research Intelligent Operation Network system.

		January	February	March	April	May	June	July	August	September	October	November	December	Total
Acute disease	2019	34,239	25,757	26,544	26,370	27,524	27,131	29,555	32,882	27,935	26,681	26,538	29,499	340,655
	2020	30,857	25,663	24,224	21,363	21,760	23,247	25,619	30,656	24,781	24,418	23,563	24,351	300,502
	IRR (95% CI)	0.90 (0.89–0.92)	1.00 (0.98–1.01)	0.91 (0.90–0.93)	0.81 (0.80–0.82)	0.79 (0.78–0.80)	0.86 (0.84–0.87)	0.87 (0.85–0.88)	0.93 (0.92–0.95)	0.89 (0.87–0.90)	0.92 (0.90–0.93)	0.89 (0.87–0.90)	0.83 (0.81–0.84)	0.88 (0.88–0.89)
	*p*-value	0.00	0.68	0.00	0.00	0.00	0.00	0.00	0.00	0.00	0.00	0.00	0.00	0.00
Disease and injury due to sport	2019	135	166	232	232	252	281	289	295	309	227	213	194	2825
	2020	141	144	51	23	17	76	146	282	225	192	194	113	1604
	IRR (95% CI)	1.04 (0.82–1.33)	0.87 (0.69–1.09)	0.22 (0.16–0.30)	0.10 (0.06–0.15)	0.07 (0.04–0.11)	0.27 (0.21–0.35)	0.51 (0.41–0.62)	0.96 (0.81–1.13)	0.73 (0.61–0.87)	0.85 (0.69–1.03)	0.91 (0.75–1.11)	0.58 (0.46–0.74)	0.57 (0.53–0.60)
	*p*-value	0.72	0.21	0.00	0.00	0.00	0.00	0.00	0.59	0.00	0.09	0.35	0.00	0.00
Fire accident	2019	58	37	40	34	33	21	38	26	35	29	25	36	412
	2020	52	37	28	22	29	18	24	31	12	26	26	48	353
	IRR (95% CI)	0.90 (0.60–1.33)	1.00 (0.62–1.62)	0.70 (0.42–1.16)	0.65 (0.36–1.14)	0.88 (0.51–1.49)	0.86 (0.43–1.69)	0.63 (0.36–1.08)	1.19 (0.69–2.09)	0.34 (0.16–0.68)	0.90 (0.51–1.58)	1.04 (0.58–1.88)	1.33 (0.85–2.11)	0.86 (0.74–0.99)
	*p*-value	0.57	1.00	0.15	0.11	0.61	0.64	0.08	0.51	0.00	0.69	0.89	0.19	0.03
Injury, poisoning, and disease due to industrial accident	2019	348	321	370	365	374	385	497	542	455	406	370	365	4798
	2020	279	317	274	282	253	349	344	504	342	368	316	305	3933
	IRR (95% CI)	0.80 (0.68–0.94)	0.99 (0.84–1.16)	0.74 (0.63–0.87)	0.77 (0.66–0.90)	0.68 (0.57–0.80)	0.91 (0.78–1.05)	0.69 (0.60–0.80)	0.93 (0.82–1.05)	0.75 (0.65–0.87)	0.91 (0.78–1.05)	0.85 (0.73–1.00)	0.84 (0.72–0.98)	0.82 (0.79–0.86)
	*p*-value	0.01	0.87	0.00	0.00	0.00	0.18	0.00	0.24	0.00	0.17	0.04	0.02	0.00
Interhospital transport	2019	2897	2445	2626	2732	2553	2492	2662	2560	2493	2581	2601	2855	31,497
	2020	2895	2451	2367	1924	1959	1996	2395	2424	2282	2493	2533	2615	28,334
	IRR (95% CI)	1.00 (0.95–1.05)	1.00 (0.95–1.06)	0.90 (0.85–0.95)	0.70 (0.66–0.75)	0.77 (0.72–0.81)	0.80 (0.75–0.85)	0.90 (0.85–0.95)	0.95 (0.90–1.00)	0.92 (0.86–0.97)	0.97 (0.91–1.02)	0.97 (0.92–1.03)	0.92 (0.87–0.97)	0.90 (0.89–0.91)
	*p*-value	0.98	0.93	0.00	0.00	0.00	0.00	0.00	0.05	0.00	0.22	0.34	0.00	0.00
Natural disaster	2019	0	0	0	0	0	3	2	1	0	4	0	0	10
	2020	8	0	0	0	0	1	2	0	0	2	0	0	13
	IRR (95% CI)	NA	NA	NA	NA	NA	0.33 (0.01–4.15)	1.00 (0.07–13.80)	NA	NA	0.50 (0.05–3.49)	NA	NA	1.30 (0.53–3.31)
	*p*-value						0.38	1.00			0.45			0.54
Other injury	2019	7116	5753	6317	6400	6157	5891	6312	6518	6253	6800	6785	7516	77,818
	2020	6936	6151	5925	5021	5237	5536	6037	5837	5752	6645	6133	6552	71,762
	IRR (95% CI)	0.97 (0.94–1.01)	1.07 (1.03–1.11)	0.94 (0.91–0.97)	0.78 (0.76–0.81)	0.85 (0.82–0.88)	0.94 (0.91–0.98)	0.96 (0.92–0.99)	0.90 (0.86–0.93)	0.92 (0.89–0.95)	0.98 (0.94–1.01)	0.90 (0.87–0.94)	0.87 (0.84–0.90)	0.92 (0.91–0.93)
	*p*-value	0.13	0.00	0.00	0.00	0.00	0.00	0.01	0.00	0.00	0.18	0.00	0.00	0.00
Self-induced injury	2019	197	195	245	216	254	291	286	270	254	258	240	247	2953
	2020	265	217	250	184	253	270	315	267	316	297	204	229	3067
	IRR (95% CI)	1.35 (1.11–1.63)	1.11 (0.91–1.36)	1.02 (0.85–1.22)	0.85 (0.70–1.04)	1.00 (0.83–1.19)	0.93 (0.78–1.10)	1.10 (0.94–1.30)	0.99 (0.83–1.18)	1.24 (1.05–1.47)	1.15 (0.97–1.37)	0.85 (0.70–1.03)	0.93 (0.77–1.11)	1.04 (0.99–1.09)
	*p*-value	0.00	0.28	0.82	0.11	0.96	0.38	0.24	0.90	0.01	0.10	0.09	0.41	0.14
Traffic accident involving car, ship, or aircraft	2019	2620	2510	2997	3248	3024	2878	3198	3068	3067	3207	3223	3159	36,199
	2020	2635	2578	2679	1891	2127	2658	2843	2695	2678	2820	2712	2818	31,134
	IRR (95% CI)	1.01 (0.95–1.06)	1.03 (0.97–1.09)	0.89 (0.85–0.94)	0.58 (0.55–0.62)	0.70 (0.67–0.74)	0.92 (0.88–0.97)	0.89 (0.84–0.94)	0.88 (0.83–0.93)	0.87 (0.83–0.92)	0.88 (0.84–0.93)	0.84 (0.80–0.89)	0.89 (0.85–0.94)	0.86 (0.85–0.87)
	*p*-value	0.84	0.34	0.00	0.00	0.00	0.00	0.00	0.00	0.00	0.00	0.00	0.00	0.00
Trauma due to assault	2019	268	207	232	232	224	228	226	256	225	217	229	252	2796
	2020	250	225	229	171	197	210	218	185	197	202	185	205	2474
	IRR (95% CI)	0.93 (0.78–1.11)	1.09 (0.90–1.32)	0.99 (0.82–1.19)	0.74 (0.60–0.90)	0.88 (0.72–1.07)	0.92 (0.76–1.12)	0.96 (0.80–1.17)	0.72 (0.59–0.88)	0.88 (0.72–1.06)	0.93 (0.76–1.13)	0.81 (0.66–0.98)	0.81 (0.67–0.98)	0.88 (0.84–0.93)
	*p*-value	0.43	0.39	0.89	0.00	0.19	0.39	0.70	0.00	0.17	0.46	0.03	0.03	0.00
Water accident	2019	5	3	6	2	2	2	7	9	9	3	1	3	52
	2020	3	4	2	6	3	5	4	2	4	5	2	3	43
	IRR (95% CI)	0.60 (0.09–3.08)	1.33 (0.23–9.10)	0.33 (0.03–1.86)	3.00 (0.54–30.39)	1.50 (0.17–17.96)	2.50 (0.41–26.25)	0.57 (0.12–2.25)	0.22 (0.02–1.07)	0.44 (0.10–1.59)	1.67 (0.32–10.73)	2.00 (0.10–117.99)	1.00 (0.13–7.47)	0.83 (0.54–1.26)
	*p*-value	0.51	0.73	0.18	0.18	0.69	0.29	0.39	0.04	0.18	0.51	0.63	1.00	0.36
Other	2019	14	9	13	11	13	12	11	7	11	7	11	60	179
	2020	9	6	9	11	9	5	8	15	4	11	5	10	102
	IRR (95% CI)	0.64 (0.25–1.59)	0.67 (0.20–2.10)	0.69 (0.26–1.75)	1.00 (0.39–2.54)	0.69 (0.26–1.75)	0.42 (0.11–1.27)	0.73 (0.25–1.99)	2.14 (0.82–6.21)	0.36 (0.08–1.23)	1.57 (0.56–4.78)	0.45 (0.12–1.42)	0.17 (0.08–0.33)	0.57 (0.44–0.73)
	*p*-value	0.31	0.45	0.40	1.00	0.40	0.10	0.50	0.09	0.08	0.36	0.14	0.00	0.00

IRR: incident rate ratio; CI: confidence interval; NA: no assessment. IRR is for 2020 versus 2019.

**Table 2 jcm-10-05662-t002:** The number of emergency patients registered in the Osaka Emergency Information Research Intelligent Operation Network system.

		January	February	March	April	May	June	July	August	September	October	November	December	Total
Total	2019	47,897	37,403	39,622	39,842	40,410	39,615	43,083	46,434	41,046	40,420	40,236	44,186	500,194
	2020	44,330	37,793	36,038	30,898	31,844	34,371	37,955	42,898	36,593	37,479	35,873	37,249	443,321
	IRR (95% CI)	0.93 (0.91–0.94)	1.01 (1.00–1.03)	0.91 (0.90–0.92)	0.78 (0.76–0.79)	0.79 (0.78–0.80)	0.87 (0.86–0.88)	0.88 (0.87–0.89)	0.92 (0.91–0.94)	0.89 (0.88–0.90)	0.93 (0.91–0.94)	0.89 (0.88–0.90)	0.84 (0.83–0.85)	0.89 (0.88–0.89)
	*p*-value	0.00	0.15	0.00	0.00	0.00	0.00	0.00	0.00	0.00	0.00	0.00	0.00	0.00
Children	2019	5108	3603	3937	4406	4565	4817	4833	4516	4269	3883	3699	4429	52,065
	2020	4199	3215	2766	2267	2293	2686	3186	3286	2949	3081	2945	2661	35,534
	IRR (95% CI)	0.82 (0.79–0.86)	0.89 (0.85–0.94)	0.70 (0.67–0.74)	0.51 (0.49–0.54)	0.50 (0.48–0.53)	0.56 (0.53–0.58)	0.66 (0.63–0.69)	0.73 (0.70–0.76)	0.69 (0.66–0.72)	0.79 (0.76–0.83)	0.80 (0.76–0.84)	0.60 (0.57–0.63)	0.68 (0.67–0.69)
	*p*-value	0.00	0.00	0.00	0.00	0.00	0.00	0.00	0.00	0.00	0.00	0.00	0.00	0.00
Adults	2019	13,925	11,519	12,824	12,782	13,116	13,142	14,689	16,034	13,762	13,364	12,478	13,890	161,525
	2020	13,441	11,635	11,647	10,034	10,534	11,623	13,243	14,640	11,948	11,891	10,890	10,683	142,209
	IRR (95% CI)	0.97 (0.94–0.99)	1.01 (0.98–1.04)	0.91 (0.89–0.93)	0.79 (0.76–0.81)	0.80 (0.78–0.82)	0.88 (0.86–0.91)	0.90 (0.88–0.92)	0.91 (0.89–0.93)	0.87 (0.85–0.89)	0.89 (0.87–0.91)	0.87 (0.85–0.90)	0.77 (0.75–0.79)	0.88 (0.87–0.89)
	*p*-value	0.00	0.45	0.00	0.00	0.00	0.00	0.00	0.00	0.00	0.00	0.00	0.00	0.00
Elderlies	2019	28,864	22,281	22,861	22,654	22,729	21,656	23,561	25,884	23,015	23,173	24,059	25,867	286,604
	2020	26,690	22,943	21,625	18,597	19,017	20,062	21,526	24,972	21,696	22,507	22,038	23,905	265,578
	IRR (95% CI)	0.92 (0.91–0.94)	1.03 (1.01–1.05)	0.95 (0.93–0.96)	0.82 (0.81–0.84)	0.84 (0.82–0.85)	0.93 (0.91–0.94)	0.91 (0.90–0.93)	0.96 (0.95–0.98)	0.94 (0.93–0.96)	0.97 (0.95–0.99)	0.92 (0.90–0.93)	0.92 (0.91–0.94)	0.93 (0.92–0.93)
	*p*-value	0.00	0.00	0.00	0.00	0.00	0.00	0.00	0.00	0.00	0.00	0.00	0.00	0.00

IRR: incident rate ratio; CI: confidence interval; NA: no assessment. IRR is for 2020 versus 2019.

**Table 3 jcm-10-05662-t003:** The number of deaths among hospitalized emergency patients registered in the Osaka Emergency Information Research Intelligent Operation Network system.

Reason for Ambulance Call		January	February	March	April	May	June	July	August	September	October	November	December	Total
Acute disease	2019	1112	829	870	770	767	670	715	698	755	791	908	942	9827
	2020	1028	913	882	756	748	695	718	723	706	800	873	1014	9856
	IRR (95% CI)	0.92 (0.85–1.01)	1.10 (1.00–1.21)	1.01 (0.92–1.11)	0.98 (0.89–1.09)	0.98 (0.88–1.08)	1.04 (0.93–1.16)	1.00 (0.90–1.12)	1.04 (0.93–1.15)	0.94 (0.84–1.04)	1.01 (0.92–1.12)	0.96 (0.88–1.06)	1.08 (0.98–1.18)	1.00 (0.98–1.03)
	*p*-value	0.07	0.04	0.77	0.72	0.63	0.50	0.94	0.51	0.20	0.82	0.41	0.10	0.84
Disease and injury due to sport	2019	0	1	0	0	0	0	0	0	0	0	0	0	1
	2020	0	0	0	0	0	0	0	0	0	0	0	0	0
	IRR (95% CI)	NA	NA	NA	NA	NA	NA	NA	NA	NA	NA	NA	NA	NA
	*p*-value													
Fire accident	2019	3	1	0	2	2	0	5	0	2	3	1	0	19
	2020	3	2	1	0	1	0	1	0	0	1	0	0	9
	IRR (95% CI)	1.00 (0.13–7.47)	2.00 (0.10–117.99)	NA	NA	0.50 (0.01–9.60)	NA	0.20 (0.00–1.79)	NA	NA	0.33 (0.01–4.15)	NA	NA	0.47 (0.19–1.10)
	*p*-value	1.00	0.63			0.63		0.13			0.38			0.06
Injury, poisoning, and disease due to industrial accident	2019	2	0	1	0	3	2	3	2	1	2	1	2	19
	2020	3	1	0	4	0	2	1	0	2	3	0	0	16
	IRR (95% CI)	1.50 (0.17–17.96)	NA	NA	NA	NA	1.00 (0.07–13.80)	0.33 (0.01–4.15)	NA	2.00 (0.10–117.99)	1.50 (0.17–17.96)	NA	NA	0.84 (0.41–1.73)
	*p*-value	0.69					1.00	0.38		0.63	0.69			0.62
Interhospital transport	2019	119	117	86	110	98	76	105	91	86	101	106	120	1215
	2020	138	92	104	100	93	80	87	124	100	114	120	148	1300
	IRR (95% CI)	1.16 (0.90–1.49)	0.79 (0.59–1.04)	1.21 (0.90–1.63)	0.91 (0.69–1.20)	0.95 (0.71–1.27)	1.05 (0.76–1.46)	0.83 (0.62–1.11)	1.36 (1.03–1.81)	1.16 (0.86–1.57)	1.13 (0.86–1.49)	1.13 (0.86–1.48)	1.23 (0.96–1.58)	1.07 (0.99–1.16)
	*p*-value	0.24	0.08	0.19	0.49	0.72	0.75	0.19	0.02	0.31	0.38	0.35	0.09	0.09
Natural disaster	2019	0	0	0	0	0	0	0	0	0	0	0	0	0
	2020	0	0	0	0	0	0	0	0	0	0	0	0	0
	IRR (95% CI)	NA	NA	NA	NA	NA	NA	NA	NA	NA	NA	NA	NA	NA
	*p*-value													
Other injury	2019	73	57	33	50	36	39	47	35	30	53	58	72	583
	2020	62	42	47	37	36	44	42	43	41	39	44	56	533
	IRR (95% CI)	0.85 (0.60–1.21)	0.74 (0.48–1.12)	1.42 (0.89–2.29)	0.74 (0.47–1.15)	1.00 (0.61–1.63)	1.13 (0.72–1.78)	0.89 (0.58–1.38)	1.23 (0.77–1.98)	1.37 (0.83–2.27)	0.74 (0.47–1.13)	0.76 (0.50–1.14)	0.78 (0.54–1.12)	0.91 (0.81–1.03)
	*p*-value	0.35	0.13	0.12	0.17	1.00	0.59	0.60	0.37	0.19	0.15	0.17	0.16	0.13
Self-induced injury	2019	8	6	7	15	13	12	11	10	5	17	12	11	127
	2020	8	10	11	8	11	9	19	15	13	14	15	11	144
	IRR (95% CI)	1.00 (0.33–3.06)	1.67 (0.55–5.58)	1.57 (0.56–4.78)	0.53 (0.20–1.34)	0.85 (0.34–2.05)	0.75 (0.28–1.94)	1.73 (0.78–4.02)	1.50 (0.63–3.73)	2.60 (0.87–9.31)	0.82 (0.38–1.78)	1.25 (0.55–2.92)	1.00 (0.39–2.54)	1.13 (0.89–1.45)
	*p*-value	1.00	0.33	0.36	0.15	0.69	0.52	0.15	0.33	0.06	0.60	0.57	1.00	0.30
Traffic accident involving car, ship, or aircraft	2019	8	7	9	11	7	7	14	10	10	14	10	15	122
	2020	9	8	13	6	7	7	1	10	9	7	9	8	94
	IRR (95% CI)	1.13 (0.39–3.35)	1.14 (0.36–3.70)	1.44 (0.57–3.83)	0.55 (0.17–1.61)	1.00 (0.30–3.34)	1.00 (0.30–3.34)	0.07 (0.00–0.47)	1.00 (0.37–2.68)	0.90 (0.32–2.46)	0.50 (0.17–1.32)	0.90 (0.32–2.46)	0.53 (0.20–1.34)	0.77 (0.58–1.02)
	*p*-value	0.81	0.80	0.40	0.24	1.00	1.00	0.00	1.00	0.82	0.13	0.82	0.15	0.06
Trauma due to assault	2019	0	0	0	2	0	1	0	1	0	1	0	0	5
	2020	0	1	0	0	0	1	1	0	0	1	0	0	4
	IRR (95% CI)	NA	NA	NA	NA	NA	1.00 (0.01–78.50)	NA	NA	NA	1.00 (0.01–78.50)	NA	NA	0.80 (0.16–3.72)
	*p*-value						1.00				1.00			0.75
Water accident	2019	0	0	0	0	1	1	0	0	1	2	0	0	5
	2020	0	0	0	1	0	0	0	0	1	0	0	0	2
	IRR (95% CI)	NA	NA	NA	NA	NA	NA	NA	NA	1.00 (0.01–78.50)	NA	NA	NA	0.40 (0.04–2.44)
	*p*-value									1.00				0.29
Other	2019	0	0	0	1	0	0	1	0	0	0	0	6	8
	2020	0	1	0	0	2	1	0	0	0	0	1	0	5
	IRR (95% CI)	NA	NA	NA	NA	NA	NA	NA	NA	NA	NA	NA	NA	0.63 (0.16–2.17)
	*p*-value													0.42

IRR: incident rate ratio; CI: confidence interval; NA: no assessment. IRR is for 2020 versus 2019.

**Table 4 jcm-10-05662-t004:** The number of deaths among hospitalized emergency patients registered in the Osaka Emergency Information Research Intelligent Operation Network system.

		January	February	March	April	May	June	July	August	September	October	November	December	Total
Total	2019	1325	1018	1006	961	927	808	901	847	890	984	1096	1168	11,931
	2020	1251	1070	1058	912	898	839	870	915	872	979	1062	1237	11,963
	IRR (95% CI)	0.94 (0.87–1.02)	1.05 (0.96–1.15)	1.05 (0.96–1.15)	0.95 (0.87–1.04)	0.97 (0.88–1.06)	1.04 (0.94–1.15)	0.97 (0.88–1.06)	1.08 (0.98–1.19)	0.98 (0.89–1.08)	0.99 (0.91–1.09)	0.97 (0.89–1.06)	1.06 (0.98–1.15)	1.00 (0.98–1.03)
	*p*-value	0.14	0.26	0.25	0.26	0.50	0.45	0.46	0.11	0.67	0.91	0.46	0.16	0.84
Children	2019	9	2	4	7	3	5	8	5	5	4	3	3	58
	2020	5	8	4	3	1	2	3	2	3	4	2	10	47
	IRR (95% CI)	0.56 (0.15–1.85)	4.00 (0.80–38.67)	1.00 (0.19–5.37)	0.43 (0.07–1.88)	0.33 (0.01–4.15)	0.40 (0.04–2.44)	0.38 (0.06–1.56)	0.40 (0.04–2.44)	0.60 (0.09–3.08)	1.00 (0.19–5.37)	0.67 (0.06–5.82)	3.33 (0.86–18.85)	0.81 (0.54–1.21)
	*p*-value	0.30	0.07	1.00	0.23	0.38	0.29	0.15	0.29	0.51	1.00	0.69	0.06	0.29
Adults	2019	173	115	123	122	110	105	119	108	107	146	149	165	1542
	2020	156	113	115	126	94	112	139	132	110	136	133	144	1510
	IRR (95% CI)	0.90 (0.72–1.13)	0.98 (0.75–1.29)	0.93 (0.72–1.22)	1.03 (0.80–1.34)	0.85 (0.64–1.14)	1.07 (0.81–1.41)	1.17 (0.91–1.50)	1.22 (0.94–1.59)	1.03 (0.78–1.35)	0.93 (0.73–1.18)	0.89 (0.70–1.14)	0.87 (0.69–1.10)	0.98 (0.91–1.05)
	*p*-value	0.35	0.89	0.60	0.80	0.26	0.64	0.21	0.12	0.84	0.55	0.34	0.23	0.56
Elderlies	2019	1143	901	879	832	814	698	774	734	778	834	944	1000	10,331
	2020	1090	949	939	783	803	725	728	781	759	839	927	1083	10,406
	IRR (95% CI)	0.95 (0.88–1.04)	1.05 (0.96–1.16)	1.07 (0.97–1.17)	0.94 (0.85–1.04)	0.99 (0.89–1.09)	1.04 (0.93–1.15)	0.94 (0.85–1.04)	1.06 (0.96–1.18)	0.98 (0.88–1.08)	1.01 (0.91–1.11)	0.98 (0.90–1.08)	1.08 (0.99–1.18)	1.01 (0.98–1.04)
	*p*-value	0.26	0.26	0.16	0.22	0.78	0.47	0.24	0.23	0.63	0.90	0.69	0.07	0.60

IRR: incident rate ratio; CI: confidence interval; NA: no assessment. IRR is for 2020 versus 2019.

**Table 5 jcm-10-05662-t005:** The number of emergency patients for acute disease registered in the Osaka Emergency Information Research Intelligent Operation Network system.

Acute Disease		January	February	March	April	May	June	July	August	September	October	November	December	Total
Children	2019	3629	2273	2219	2451	2592	2924	2892	2776	2395	2089	1948	2773	30,961
	2020	2837	1971	1500	1161	1027	1321	1662	1816	1426	1463	1411	1334	18,929
	IRR (95% CI)	0.78 (0.74–0.82)	0.87 (0.82–0.92)	0.68 (0.63–0.72)	0.47 (0.44–0.51)	0.40 (0.37–0.43)	0.45 (0.42–0.48)	0.57 (0.54–0.61)	0.65 (0.62–0.69)	0.60 (0.56–0.64)	0.70 (0.65–0.75)	0.72 (0.68–0.78)	0.48 (0.45–0.51)	0.61 (0.60–0.62)
Adults	2019	9748	7644	8368	8266	8718	8792	9898	11,180	9155	8649	8083	9133	107,634
	2020	9235	7669	7633	7025	7233	7781	8917	10,421	7999	7586	7088	6768	95,355
	IRR (95% CI)	0.95 (0.92–0.97)	1.00 (0.97–1.04)	0.91 (0.88–0.94)	0.85 (0.82–0.88)	0.83 (0.80–0.86)	0.89 (0.86–0.91)	0.90 (0.88–0.93)	0.93 (0.91–0.96)	0.87 (0.85–0.90)	0.88 (0.85–0.90)	0.88 (0.85–0.91)	0.74 (0.72–0.76)	0.89 (0.88–0.89)
Elderlies	2019	20,862	15,840	15,957	15,653	16,214	15,415	16,765	18,926	16,385	15,943	16,507	17,593	202,060
	2020	18,785	16,023	15,091	13,177	13,500	14,145	15,040	18,419	15,356	15,369	15,064	16,249	186,218
	IRR (95% CI)	0.90 (0.88–0.92)	1.01 (0.99–1.03)	0.95 (0.92–0.97)	0.84 (0.82–0.86)	0.83 (0.81–0.85)	0.92 (0.90–0.94)	0.90 (0.88–0.92)	0.97 (0.95–0.99)	0.94 (0.92–0.96)	0.96 (0.94–0.99)	0.91 (0.89–0.93)	0.92 (0.90–0.94)	0.92 (0.92–0.93)

IRR: incident rate ratio; CI: confidence interval; NA: not assessment.

**Table 6 jcm-10-05662-t006:** The number of deaths among hospitalized emergency patients for acute disease registered in the Osaka Emergency Information Research Intelligent Operation Network system.

Reason for Ambulance Call		January	February	March	April	May	June	July	August	September	October	November	December	Total
Children	2019	4	2	1	2	2	3	3	2	3	2	0	2	26
	2020	4	2	2	2	1	2	2	1	1	1	0	7	25
	IRR (95% CI)	1.00 (0.19–5.37)	1.00 (0.07–13.80)	2.00 (0.10–117.99)	1.00 (0.07–13.80)	0.50 (0.01–9.60)	0.67 (0.06–5.82)	0.67 (0.06–5.82)	0.50 (0.01–9.60)	0.33 (0.01–4.15)	0.50 (0.01–9.60)	NA	3.50 (0.67–34.53)	0.96 (0.53–1.73)
Adults	2019	143	84	107	96	79	88	88	81	92	106	117	129	1210
	2020	124	90	88	95	75	90	108	100	84	95	106	116	1171
	IRR (95% CI)	0.87 (0.68–1.11)	1.07 (0.79–1.46)	0.82 (0.61–1.10)	0.99 (0.74–1.33)	0.95 (0.68–1.32)	1.02 (0.75–1.39)	1.23 (0.92–1.65)	1.23 (0.91–1.68)	0.91 (0.67–1.24)	0.90 (0.67–1.19)	0.91 (0.69–1.19)	0.90 (0.69–1.16)	0.97 (0.89–1.05)
Elderlies	2019	965	743	762	672	686	579	624	615	660	683	791	811	8591
	2020	900	821	792	659	672	603	608	622	621	704	767	891	8660
	IRR (95% CI)	0.93 (0.85–1.02)	1.10 (1.00–1.22)	1.04 (0.94–1.15)	0.98 (0.88–1.09)	0.98 (0.88–1.09)	1.04 (0.93–1.17)	0.97 (0.87–1.09)	1.01 (0.90–1.13)	0.94 (0.84–1.05)	1.03 (0.93–1.15)	0.97 (0.88–1.07)	1.10 (1.00–1.21)	1.01 (0.98–1.04)

IRR: incident rate ratio; CI: confidence interval; NA: not assessment.

## Data Availability

The data that support the findings of this study are available from the Osaka Prefectural government, but the availability of these data is restricted. Data cannot be shared publicly because of the Protection Ordinance for Personal Information in Osaka Prefecture. Data may be applied for if a qualified researcher applies for the data and the research is approved by the technical committee (http://www.pref.osaka.lg.jp/iryo/qq/orion_teikyo.html, (accessed on 1 November 2021), in Japanese).

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
