# Peer review of "Incidence and Mortality of Emergency Patients Transported by Emergency Medical Service Personnel during the Novel Corona Virus Pandemic in Osaka Prefecture, Japan: A Population-Based Study"

_jcm, 2021, doi:10.3390/jcm10235662_

Round 1
Reviewer 1 Report
Incidence and mortality of emergency patients transported by emergency medical service personnel during the novel corona virus pandemic in Osaka Prefecture, Japan: A population-based study
The aim of the study was “to determine the impact of the COVID-19 pandemic on the incident number of emergency patients transported by ambulance and the number of deaths among emergency patients admitted to hospital”.
For this goal, the Authors use data collected by the ORION system, an informative system integrating data of ambulance calls with hospital records. IRRs, calculated with a Poisson regression model, show that the incidence of ambulance calls decreased in 2020 compared to 2019, but the mortality of transported patients did not significantly change. These indicators were calculated by month, age, gender and reason for ambulance call.
General comment
This interesting study provides indications on how the COVID-19 pandemic affected some aspects of the health care system by using a very informative and large database. This is an example of how data routinely collected can provide useful information. However some clarifications are needed and some points should be discussed.
The indicators based on the incident number of ambulance calls by reason is informative, although there are some things to be clarified about the model used. The information derived from the mortality is also useful, but has some important limitations that should be discussed. This indicator uses the number of deaths of emergency patients (transported and hospitalized) referred to the whole population. This is only one part of the overall mortality of the population. With this indicator authors cannot take into account of the possible increase of mortality outside the observational field.
In consideration of the above, the declared objectives are reached in the paper, but some caution should be used in data interpretation, although the general conclusion could be correct.
The discussion (also some parts of the introduction) could be synthesized in some part (especially the paragraph 227-258) not essential to the article. In general, the article would benefit of some minor revisions of the writing.
Looking forward to see analysis by ICD codes.
Specific comments
Table 1 and 2 are useful but not legible. I suggest to move these tables in the supplementary material and to provide in the main test only the essential information (for instance reasons for call with few cases can be omitted, confidence intervals of IRR can be substituted by an indication of significance, the part by age group could be separated in different tables). This partially applies also for table 3 and 4.
Line 77-78: The expression “emergency patients” is often used. Does this expression correspond to “patients who were transported by ambulance”? You could specify this in parenthesis to help the reader: “patients who were transported by ambulance (emergency patients)”.
Line 90-91: It is not very clear how many people have been excluded from the analysis (percent of exclusion in both years). (see also comment for line 114-127)
Line 93-106 section Exposure: this paragraph is not essential here. It can be misleading because the COVID-19 status of the patients has not been used in data analysis. The authors could report this part (or a synthesis of it) in the discussion, since it is useful to help the discussion of the results.
Line 111-112: Reason for the exclusion of deaths in the emergency department is not provided. Are they excluded because the patient’s outcome is not known when there is not admission to hospital? Could this be a limitation of the study to be discussed?
Line 114-127 section Measurement: it would be important to have an idea of how many records have been excluded from the analysis because of inconsistent/incomplete data. (Has the quality of data improved during the years?)
Line 139-140: Are the IRRs calculated for all ages adjusted by age and sex? Are there other adjustment variables in the model? If so, this should be specified.
Line 175-178: Not very clear: in children the decrease is bigger, but the decrease is significant also in adults and elderly for nearly all months, with the exclusion of February.
Line 287-286: The indicator is calculated only on a part of the mortality but results are extended to the whole population. Some bias can affect this indicator. Since the ambulance calls decreased, it is also possible that there was an increase in mortality of non-transported people. Caution should be expressed in the discussion of the results. Is there any finding in the literature (on excess mortality or even causes of death) that could confirm or support the comments of the authors? For instance here for the whole Japan Yorifuji T, Matsumoto N, Takao S. Excess All-Cause Mortality During the COVID-19 Outbreak in Japan. J Epidemiol. 2021 Jan 5;31(1):90-92. doi: 10.2188/jea.JE20200492. Epub 2020 Nov 25. PMID: 33132284; PMCID: PMC7738637.
(not necessary to address in the paper) The number of emergency patients decreased between 2019 and 2020 but the deaths stayed the same. Is this an indication of an increase of risk of death for the transported patients? Have the authors a reason for this increased risk for the transported patients? Would such indicator be interesting for further studies?
Other minor things:
Line 30-31: The sentence is not completely clear, in particular what the author mean by “impact”.
Line n. 49: “In Japan, the number of patients with COVID-19 about 740,000 on 31 May 2021 [1”]. Should it be “In Japan, the number of patients with COVID-19 was about 740,000 on 31 May 2021 [1].”?
Line 53: a reference could be provided.
Line 70-73: The expression “number of deaths among emergency patients” gives the impression that the paper provides a measure of mortality risk for emergency patients. In fact, the mortality indicator has been referred to the whole population not to the transported patients. Maybe it could be rephrased as “number of deaths of emergency patients”.
Line 77-79: The sentence is not completely clear. Is it a description of the data included in the ORION system (“data are entered in the system”) or is it a description of data used in the study (maybe is missing “were used” at the endo of the sentence…)?
Line n. 132: the control year is 2019?
Line n. 158: the number 203,89 is not correct.
Line n. 214-216: Some leftover text is present.
Author Response
Thank you for your thorough reviews and important suggestions. Please see the attachment about our responses to your queries.

Reviewer 2 Report
Dear Authors,
I’ve read your manuscript and I founded very interesting. Related data about your experience in extra hospital emergency medical service are very impressive, even you can’t estimate the real significance of missing patients in ORION database.
Before editing I wonder if you could better correlate these results with a general clinical developement of corona-virus infection, not only to the specific japanese features of first pandemic outbreak.
First, evem considering the impressive level of your database, you don’t report the rate of the rate of acute respiratory failure as reason for ambulance call, but only a more generic “Acute disease”; so the real impact of Covid-19 infection is more difficult to understand and evaluate.
Moreover: Line 52-54 you should correct with” 20% of Covid-19 patients are severely affected and hospital admission, and a lower but not negligible rate (3-4%) also intensive management in ICU for their acute respiratory failure by intubation and mechanical ventilation.
Finally I’d appreciate to know if you can report or exclude a significative number of emergency medical service staff’s viral infections, explaining or excluding critical issues in your emergency medical system for extra hospital taking care of the patients.
Best regards
Author Response

(The authors gave the same response as above.)
